# Molecular Epidemiology Reveals the Co-Circulation of Two Genotypes of Coxsackievirus B5 in China

**DOI:** 10.3390/v14122693

**Published:** 2022-11-30

**Authors:** Yun He, Haiyan Wei, Leilei Wei, Huan Fan, Dongmei Yan, Hua Zhao, Shuangli Zhu, Tianjiao Ji, Jinbo Xiao, Huanhuan Lu, Wenhui Wang, Qin Guo, Qian Yang, Weijia Xing, Yong Zhang

**Affiliations:** 1National Polio Laboratory, WHO WPRO Regional Polio Reference Laboratory, National Health Commission Key Laboratory for Biosafety, National Health Commission Key Laboratory for Medical Virology, National Institute for Viral Disease Control and Prevention, Chinese Center for Disease Control and Prevention, Beijing 102206, China; 2School of Public Health and Management, Shandong First Medical University, Shandong Academy of Medical Sciences, 6699 Qindao Road, Jinan 250117, China; 3Henan Center for Disease Control and Prevention, Zhengzhou 450016, China; 4Jilin Center for Disease Control and Prevention, Jilin Institute of Public Health, Changchun 130062, China; 5Jiangsu Center for Disease Control and Prevention, Nanjing 210009, China; 6Chongqing Center for Disease Control and Prevention, Chongqing 400042, China; 7Center for Biosafety Mega-Science, Chinese Academy of Sciences, Wuhan 430071, China

**Keywords:** coxsackievirus B5, genotyping, evolutionary dynamics, recombination analysis, genetic diversity, hand, foot, mouth disease

## Abstract

Coxsackievirus B5 (CVB5) is an important enterovirus B species (EV-Bs) type. We used the full-length genomic sequences of 53 viral sequences from the national hand, foot, and mouth disease surveillance network in the Chinese mainland (2001–2021). Among them, 69 entire *VP1* coding region nucleotide sequences were used for CVB5 genotyping and genetic evolution analysis. Phylogenetic analysis based on a data set of 448 complete VP1 sequences showed that CVB5 could be divided into four genotypes (A-D) worldwide. Sequences from this study belonged to genotypes B and D, which dominated transmission in the Chinese mainland. Two transmission lineages of CVB5 have been discovered in the Chinese mainland, lineage 2 was predominant. Markov chain Monte Carlo analysis indicated that the tMRCA of CVB5 in the Chinese mainland could be traced to 1955, while the global trend could be traced to 1862, 93 years earlier than China. The evolution rate of CVB5 was higher in the Chinese mainland than worldwide. The spatiotemporal dynamics analysis of CVB5 assessed that virus transportation events were relatively active in Central, Northeast, North and Northwest China. Recombination analysis revealed frequent intertypic recombination in the non-structural region of CVB5 genotypes B and D with the other EV-Bs, revealing eight recombination lineages. Our study showed the molecular evolution and phylogeography of CVB5 that could provide valuable information for disease prevention.

## 1. Introduction

Human enteroviruses (EVs) belong to the genus Enterovirus of the family *Picornaviridae*, which has been classified under EV-A-D [1], including Coxsackievirus A and B (CVA and CVB), Poliovirus, Echoviruses, and newly identified EVs [2]. Coxsackievirus B5 (CVB5) is one of the most common types of the CVB group (types 1–6). It causes various clinical diseases, such as aseptic meningitis (AM) [3,4]; viral encephalitis (VE) [5,6]; hand, foot, and mouth disease (HFMD) [7,8]; acute flaccid paralysis (AFP) [9]; type 1 diabetes [10]; and other diseases, which seriously threaten the health of infants and young children.

The prototype Faulkner strain of CVB5 was first isolated from the USA in 1952. Since 1957, CVB5 has repeatedly been reported worldwide to cause diseases or outbreaks, such as AM, encephalitis, myocarditis, HFMD, and pericarditis [3,5,10]. In China, CVB5-related HFMD and AM cases have been reported repeatedly in many provinces [11,12]; in recent years, the proportion of CVB5 infections increased in HFMD cases [13]. In addition, periodic epidemics of CVB5 can be accompanied by the evolution of some strains resulting in enhanced virulence, and CVB5-infected HFMD cases can cause neurological manifestations, which could even lead to severe cases and death [7,14]. Consequently, enhanced surveillance of CVB5 is critical for disease prevention and control.

There have been some relevant studies on the genomic epidemiology of CVB5 and global genetic evolution analysis [15]. However, due to the limitations of the selected sequences with a narrow separation time range and few regions covered, the current molecular evolution study of CVB5 could not be well explained. Therefore, in this study, we explored CVB5 gene diversity, genetic evolution, and global transmission patterns by analyzing the VP1 coding region and full-length genomic sequences of CVB5 based on data from the Chinese HFMD Surveillance Network from 2001 to 2021. We combined phylogenetics with phylogeographic analysis to clarify the evolutionary relationship and spatiotemporal dynamics of CVB5 in China. This study provides preliminary data for preventing, diagnosing, and treating CVB5-induced diseases.

## 2. Materials and Methods

### 2.1. Sample Collection

The CVB5 samples were obtained from the Chinese HFMD Surveillance Network and were screened from 2001 to 2021. Two cell lines, human rhabdomyosarcoma (RD) cells and human laryngeal epithelial (Hep-2) cells, were used for virus isolation simultaneously. All samples were inoculated into each of the two cell lines and observed daily for cytopathic effect (CPE). Infected cell cultures were harvested following the appearance of complete EV-like CPE. All experimental protocols were adopted by the National Institute for Viral Disease Control and Prevention, and the methods were carried out in accordance with the approved guidelines [16]. The use and analysis of all samples in this study were approved by the Second Ethics Review Committee of the National Institute for Viral Disease Control and Prevention, Chinese Center for Disease Control and Prevention.

### 2.2. Molecular Typing and Full-Length Genome Sequencing

We used nucleic acid extraction kits (CqEx-DNA/RNA virus, Tianlong, China) to extract viral nucleic acids. In RT-PCR amplification kit ver 2 (Takara biomedical, Dalian, China), the amplification of the full-length coding region sequence of CVB5 VP1 coding region referred to primer 490/491/492/493 [17]. The amplification of the genome sequence of CVB5 adopted the segmented amplification method and referred to the genome sequence downloaded from GenBank to design the primer sequence (Table 1).

The total volume of the PCR reaction system was 25 μL, including 12.5 μL of 2× Mix buffer, 7.5 μL of RNA-free H_2_O, 1 μL of Taq mix enzyme, 0.5 μL of forward primer, 0.5 μL of reverse primer, and 3 μL of the template. The PCR procedure was divided into three stages: reverse transcription (30 min at 50 ℃, 2 min at 95 ℃), sequence amplification (30 s at 94 ℃, 40 s at 50 ℃, 90 s at 72 ℃, 40 cycles), and sequence extension (10 min at 72 ℃). The final product was stored at 4 ℃. All amplified positive products were sent to the company (Tsingke Biotechnology Co., Ltd., Beijing, China) for sequencing. The amplified positive products were purified using the QIAquick PCR Purification Kit (Qiagen, Hilden, Germany), and then the forward and reverse primers were used, respectively. The two-way labelling reaction was performed according to the labelling reaction conditions in the BigDye™ Terminator V3.1 Cycle Sequencing Ready reaction kit (Applied Biosystems-systems, Foster City, CA, USA) for bidirectional labelling reaction. The labelled product was purified using dextran gel G50 (Pharmacia, Uppsala, Sweden) and then sequenced and analyzed on the ABI 3130 Genetic Analyzer (Applied Biosystems, Foster City, CA, USA).

### 2.3. Phylogeny and Bioinformatics Analysis

The sequencher software (version 5.4.5, GeneCodes, Ann Arbor, Michigan, USA) was used to proofread and sort out the sequenced map. The Clustal W software package in MEGA software (version 11.0.11, Sudhir Kumar, Arizona State University, Tempe, Arizona, USA) was used for the multi-sequence alignment [20]. The most suitable analytical evolution model was selected using the Find Best Model program. The full-length phylogenetic tree of the VP1 coding region based on CVB5 was constructed via the maximum likelihood method (ML). The reliability of tree building is evaluated through 1000 bootstraps. The 363 VP1 sequence was screened for BEAST analysis (version 1.10, Marc A. Suchard, Los Angeles, CA, USA) [21].

### 2.4. Recombination Analysis

For the recombination analysis of the CVB5 strain, we used the sequences of EV-B prototype strains from the GenBank, and 53 CVB5 full-length genome sequences from this study were selected to construct the neighbor-joining (NJ) phylogenetic trees of 5′UTR, P1, P2, P3 region using MEGA (version 11.0) with 1000 bootstrap replicates [20]. Based on 5′UTR, P2, and P3 phylogenetic trees, a representative CVB5 strain was selected from each evolutionary clade. The Recombinant Detection Program (RDP4, version 4.46, Darren P. Martin, Cape Town, South Africa,) was used to analyze the recombination signal, and seven methods were applied, including RDP, GENECONV, 3Seq, Chimarea, SiScan, MaxChi, and Lard [22]. The BLAST sever was used to search for and download sequences with more than 85% similarity on GenBank. Phylogenetic incongruence between different regions with a P value of less than 0.05 was considered compelling evidence for recombination.

### 2.5. Prediction of Amino Acid Mutation Sites of CVB5

Amino acid mutation sites for amino acid sequence logos were generated using WebLogo website, a web-based application [23] (http://weblogo.threeplusone.com/, accessed on 15 November 2022).

The ratio of non-synonymous substitutions to synonymous substitutions (dN/dS) was estimated by the Mixed Effects Model of Evolutionary (MEME) [24] on the Datamonkey website (http://www.Datamonkey.org, accessed on 17 July 2022).

## 3. Results

### 3.1. Summary of CVB5 Base Data

In total, 69 entire VP1 coding region sequences and 53 full-length genome sequences (Appendix A) were obtained from 2001 to 2021 during the HFMD surveillance in China, except for 1 AM sample from Anhui province. Of all 69 VP1 samples in this study, 1 was from a patient with AM (1.45%), 21 were from patients with severe HFMD (30.88%), and 47 were from patients with mild HFMD (69.12%). The classification criteria for HFMD cases (severe or mild) referred to the Guidelines for “Chinese guidelines for the diagnosis and treatment of hand, foot and mouth disease (2018 edition)” [25]. The VP1 sequence of CVB5 showed that the total length of VP1 was 849 bp bases and encoded 283 amino acids. Compared with the CVB5 prototype strain, the similarity of the VP1 nucleotide sequence and the amino acid sequence was 77.3%–81.6% and 93.2%–95.3%, respectively. All 53 full-length genomic sequences in this study were from the Chinese mainland of 16 provinces/municipalities distributed in six major geographical administrative regions of China: Northwest China (8, 15.38%), North China (9, 17.31%), Northeast China (7, 13.46%), Eastern China (13, 25%), Central China (14, 26.93%), and Southwest China (1, 1.92%). The length of full-length genomic sequences of CVB5 was 7360–7429nt, and compared with the CVB5 prototype strain, the similarities of the nucleotide sequence and amino acid sequence of the full-length genomic sequence were 79.2%–81.7% and 76.4%–79.2%, respectively.

### 3.2. Genotype Distribution of Global CVB5

The genotype division of CVB5 is based on the entire *VP1* coding region nucleotide sequence, and 448 CVB5 *VP1* sequences were obtained from GenBank, in addition to 69 CVB5 intact *VP1* sequences. The phylogenetic tree results showed that the CVB5 sequences could be divided into four relatively independent branches. According to the EV genotype classification principle, the sequences in the branches with an average nucleotide difference of 15%~25% of the tree were included in the same genotype [26]. Then, the global CVB5 sequences were divided into four genotypes: A, B, C, and D (Figure 1).

Genotype A included four sequences: the prototype was originally isolated in 1952 (GenBank accession Number: AF114383.1), the Ecuador sequence in 1974 (GenBank accession Number: GU300056.1), the France sequence in 1977 (GenBank accession Number: HF948028), and the Poland sequence in 2001 (GenBank accession Number: KU189238.1). Genotypes B, C, and D circulated from 1954-2021, 1970-2011, and 1982-2018, respectively. Genotypes A and C of global CVB5 might gradually disappear and become less prevalent during evolution, while genotypes B and D showed a global co-epidemic trend.

### 3.3. Phylogenetic Analysis of CVB5 in China

From the Markov jump model (Figure 2A), we observed that outward migration to South China (Guangdong Province and Hong Kong) and Northeast China was predominant. Meanwhile, inward migration was predominant in Northern China (Hebei Province, Beijing Municipality, and Tianjin Municipality) and Northeast China, followed by South China.

Two propagation lineages of CVB5 have been found in China based on the CVB5 molecular evolution tree (Figure 2B). It showed that Lineage 1 circulated in China from 1983 to 2001 and tended to disappear. The Lineage 2 strain was discovered between 2001 and 2020 and has dominated CVB5 epidemics and outbreaks in China. It can be seen from the two evolutionary clusters on the CVB5 molecular evolutionary tree in China that the earliest was in Taiwan (1983) and Shandong (1998), and then spread from Shandong to all parts of the country, forming multiple transmission chains. The MCC tree showed the rate of CVB5 diffusion in China, revealing phylogenetic associations between different strains. A Bayesian method was used to study Chinese CVB5 strains. Then, the phylogenetic relationship and propagation trend of CVB5 in China were discussed, and the time signals of multiple datasets were verified using root tip regression and BETS. All results support the presence of time signals in the dataset, with an R2 value of 0.6274 and a high Bayesian factor (BFs).

Simultaneously, through the Bayesian skyline map (Figure 2B), the evolutionary epidemiological characteristics of the *VP1* nucleotide sequence of the CVB5 in China were analyzed. It was found that the genetic diversity of VP1 in China was relatively stable between 1970 and 1990, and since then the genetic diversity of nucleotides in the VP1 region slightly fluctuated twice from 2005 to 2015 and from 2015 to 2018. This led to a significant increase in genetic diversity in the CVB5 VP1 region and eventually at a higher level of genetic variation. This is consistent with the MCC tree’s inference.

Compared with the global average evolution rate of the CVB5 VP1 region in 3.718 × 10^−3^ substitution/site/year (95% HPD range 3.28 × 10^−3^–4.15 × 10^−3^), the evolution rate of the CVB5 VP1 region in China was 4.807 × 10^−3^ substitution/site/year (95% HPD range 3.71 × 10^−3^–5.91 × 10^−3^), which was slightly higher than the global average. This may indicate that the nucleotide sequence of the Chinese CVB5 VP1 region is evolutionarily active.

The “wavy” changes in the nucleotide genetic diversity of the VP1 region of CVB5 in China may indicate the possibility of a periodic epidemic of the viruses, which in turn leads to changes in the genetic diversity of nucleotide sequences in the VP1 region and gradually stabilizes at a high genetic level. This is consistent with the literature that CVB5 had a unique circular pattern that differs from the other five types, with an epidemic occurring every 3–6 years, each lasting about 1 year [27,28,29,30]. Therefore, it is necessary to enhance the monitoring of CVB5 and related diseases.

### 3.4. Inference of the CVB5 Geographic Diffusion Pattern in China

The VP1 sequence was used to analyze the geographical distribution of CVB5 Chinese strains in seven geographical regions of China, and SpreaD3 (version 0.9.7.1) (Filip Bielejec, Leuven, Belgium) was used to describe the distribution [31]. We used BEAST to further analyze the temporal evolution of CVB5 in China using the complete VP1 sequence from 1983 to 2020 from 24 Provinces (including 17 provinces in our study), cities, and autonomous regions in China. Systematic geographic reconstruction confirmed that CVB5 originated in Zhejiang Province, East China, around 1980 (Figure 3).

It can be seen that the inflow and outflow of CVB5 are frequently in China, which is consistent with the prediction of CVB5 in China by the Markov jump model. Based on the value of BF (supported with BF ≥ 3), there are five significant migration paths: (1) North China to Central China; (2) North China to East China; (3) East China to Northwest China; (4) South China to Southwest China; (5) Northeast China to Northwest China (Appendix A). Our diffusion rates suggested that Central China, Northeast China, North China, and Northwest China play a key role in the transmission process of CVB5 in China. Meanwhile, Northeast China, South China, and Southwest China are also very important in the whole transmission network (Appendix A).

### 3.5. Recombinant Analysis of CVB5

In this study, 53 full-length genome sequences of CVB5 from the Chinese mainland were used for recombination analysis, including 6 sequences of genotype B, and 47 sequences of genotype D. NJ trees were constructed using the 5’UTR, P1 capsid region, P2 capsid region, and P3 non-capsid region of 53 CVB5 whole gene sequences combined with other EV-B prototype strains (Figure 4A–D).

Compared with the P1 region, the 5′UTR, P2 capsid, and P3 non-capsid regions showed significant differences, suggesting multilinear recombination among CVB5 strains. Moreover, the recombination within each genotype was not monophyletic. From the NJ tree of the P3 non-capsid (Figure 4D), the tree showed two independent lineages, indicating that two different recombination events occurred between strains of genotype B. Also, the differences in the D genotypes were more pronounced than those in the B genotype; 6 recombinant lineages were detected, indicating that CVB5 had different evolutionary pathways. The strains forming the recombinant lineages in the P3 region clustered most closely in the 5′UTR (Figure 4A) and *P2* (Figure 4C) region, revealing that the whole genome was not monophyletic in the lineage. Consequently, genotypes B and D contain 2 and 6 recombinant lineages, respectively; they are named recombinant lineages 1–8.

To further investigate the differences in recombination patterns between genotypes B and D, we performed the sliding window analysis using the DNASP package of the CVB5 genome sequences (Figure 5A).

When all genotypes were included in the calculations, the trend of whole genome sequences was roughly the same as that of genotype D, which may be related to the numerical advantage of genotype D. The comparison of nucleotide homology between different recombination branches showed that there were obvious differences between each recombination branch. Taking six recombination branches of the D genotype prevalent in China as an example, the similarity results showed high similarity between each recombination branch in the structural protein region. However, significant differences were found in the non-structural protein region (Figure 5B), which also proved the scientific division of the recombination branch. We used non-capsid regions 2A, 2B, 2C, 3AB, 3C, and 3D from 53 whole genome sequences. We then selected a representative strain from each lineage for BLAST comparison in the GenBank database. Then, all sequences with more than 85% homology of EV-B type were downloaded and included in recombination event analysis. Notably, genomic mapping of CVB5 representative strain recombination events was predicted using RDP4 software (Figure 5C).

### 3.6. Phylogenetic Analysis of CVB5 Globally

The maximum clade credibility (MCC) tree, constructed using 363 entire VP1 coding region sequences of global CVB5, had the homologous topological structure as its phylogenetic tree, and the classification results were similar. Time phylogeny provided a clue to the rate of evolution, and the estimated tMRCA was 1862. The mean substitution rate in the VP1 region was 3.718 × 10^−3^ substitution/site/year (95% HPD range 3.28 × 10^−3^–4.15 × 10^−3^). The MCC tree of global CVB5 was divided into two evolutionary clusters (Figure 6).

Since the genetic differentiation of the CVB5 strain appeared in 1952, it has continuously differentiated and spread quickly to other regions. It has formed a common cycle of multiple transmission chains in many countries and regions. The global CVB5 molecular evolution tree shows that CVB5 is widely spread in six continents, mainly in Europe and Asia, among which the most countries involved in the genetic evolution of CVB5 are in Europe (15 countries). It is followed by Asia (9 countries). It is worth noting that the B, C, and D genotypes were distributed with Chinese and French strains. Therefore, it can be inferred that the Chinese and French strains have a co-circulation trend with the CVB5 epidemic strains worldwide.

### 3.7. Amino Acid Sites Prediction of CVB5

Genetic pressure (dN/dS ratio) was 0.029 and 0.044 for genotypes B and D of global CVB5, respectively, indicating that mutation and evolution of the CVB5 gene are stable worldwide. And there were three negative selections (dN/ < 1) of the global CVB5 B genotype: aa1, aa52, and aa272. Meanwhile, there were nine positive selections (dN/dS > 1) of the global CVB5 D genotype: aa9, aa35, aa36, aa43, aa91 (VP1 BC loop), aa120, aa227 (VP2 HI loop), aa251 (VP2 C-terminal), and aa260 [32]. Therefore, we further and comprehensively analyzed the mutations at amino acid VP1 sites in the four genotypes of CVB5, and showed that a total of 26 amino acid sites were mutated (Figure 7). It is worth noting that aa84 and aa85 are in the BC loop, aa89, aa130, aa132 are in the DE loop.

## 4. Discussion

After EV molecular typing gradually replaced serum neutralization tests and became the gold standard for enteroviral typing, the *VP1* nucleotide coding region became the dominant region for EV typing [33,34]. Therefore, according to the difference of 15% to 25% of the nucleotide sequence observed in this study, the *VP1* nucleotide coding region sequence of global CVB5 is divided into four genotypes (A–D); this is highly significant for preventing and controlling CVB5 worldwide.

We found that the average evolutionary rate of CVB5 in the world was 3.718 × 10^−3^ substitution/site/year (95% HPD range 3.28 × 10^−3^–4.15 × 10^−3^), and the genetic pressure (dN/dS ratio) showed that the gene selection pressure of genotype D is higher than that of genotype B, both of which are less than 1, indicating that the mutation region of genotype B and genotype D is stable, and the positive selection sites on the capsid of genotype D deserves further attention.

Compared with the global average CVB5 evolution rate, the rate of CVB5 evolution in China was 4.807 × 10^−3^ substitution/site/year (95% HPD range 3.71 × 10^−3^–5.91 × 10^−3^) higher than the global evolution rate. Huang [32] also suggested that the transmission hub of CVB5 is mainly in China. Consequently, China should strengthen the prevention and monitoring of border viruses to effectively prevent the spread and epidemic of the virus throughout the country.

Previous studies have suggested that RNA genome recombination is one of the main drivers of RNA virus evolution [35]. Among EV species, HEV-B has been shown to have the highest recombination rate [36,37], especially among members of the same species. P1 and P2 capsid regions are the major regions of frequent recombination of EV genes [38]. This was also confirmed in this study; extensive genetic information exchange was conducted with other EV-B types: Echol 18, CVB3, Echol 11, Echol 14, CVB4, HEVB80, Echol 6, HEVB84, HEVB106, and Echol 1. Notably, the Heilongjiang strain (CVB5-HLJ-_137/CHN/2018) in genotype B in this study recombined with the CVB3 prototype NANCY, possibly due to the continuous evolution of the virus. Alternatively, further virological studies of CVB5, such as reverse genetic studies, are still necessary.

Because the main pathogen of HFMD is EV-A (including EV-A71, CVA16, CVA6, etc.), although EV-B is also the pathogen causing HFMD, the number of EV-B sequences is much smaller compared with EV-A. Therefore, in this study, molecular epidemiological studies were conducted based on CVB5 detected in HFMD cases, and the number was relatively small, especially in Southwest China, we only collected one CVB5, which may have an impact on the results of molecular epidemiology, especially on the analysis of the geographic diffusion pattern, due to geographical bias. We will collect more CVB5 sequences and conduct more in-depth studies by enhancing CVB5 surveillance in China. All these indicate that further surveillance of CVB5 is greatly needed.

## Figures and Tables

**Figure 1 viruses-14-02693-f001:**
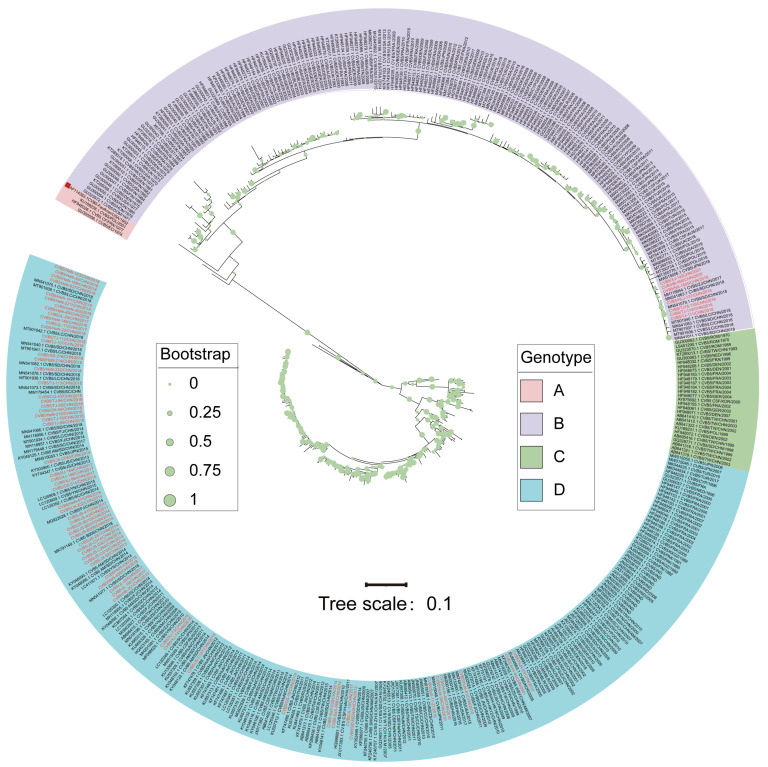
Maximum likelihood phylogenetic tree based on the entire *VP1* nucleotide coding region sequences (849bp). Notes: Regions marked in red indicate the sequences isolated in this study. The red quadrilateral represents the prototype strain.

**Figure 2 viruses-14-02693-f002:**
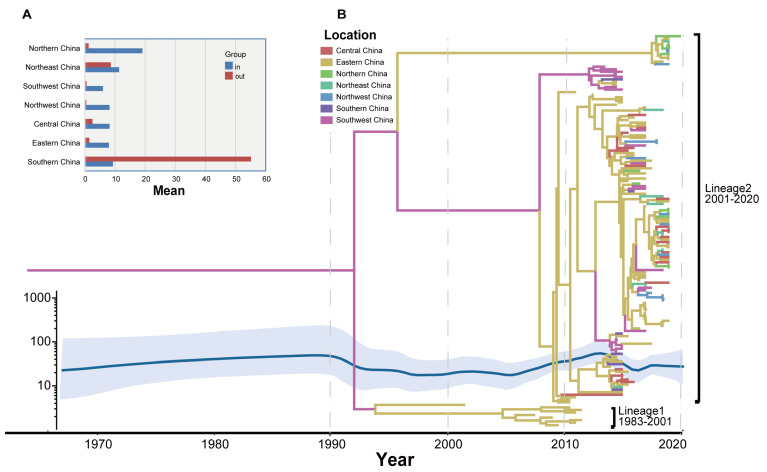
Temporal phylogenetic development of CVB5. (**A**) A histogram based on the weekly square count of different provinces. (**B**) Markov chain Monte Carlo (MCC) phylogeny tree based on the entire *VP1* coding area in China, colored according to different geographical partitions.

**Figure 3 viruses-14-02693-f003:**
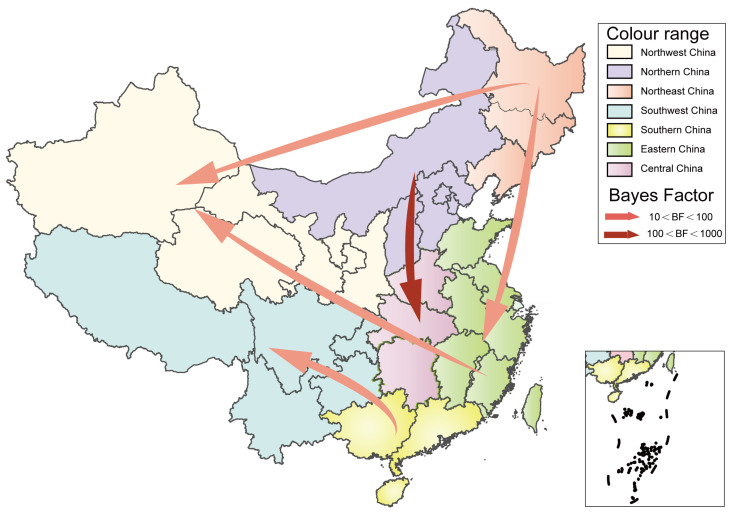
Spatiotemporal dynamics of CVB5 in China. The arrows between the locations indicate the migration route of CVB5. Only the propagation path with BF ≥ 3 is shown.

**Figure 4 viruses-14-02693-f004:**
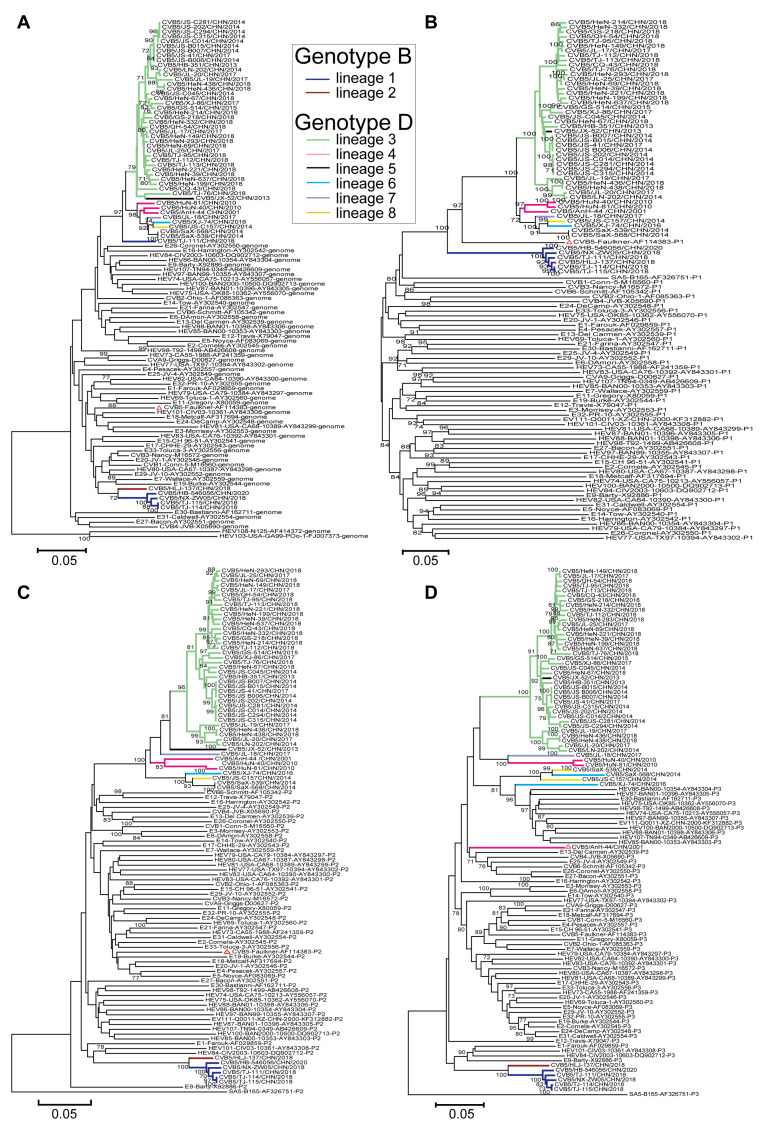
A total of 53 CVB5 isolates were analyzed with phylogenetic trees of enterovirus (EV-B) prevalent strains downloaded from GenBank. Notes: (**A**) 5′-untranslated region, (**B**) P1 capsid region, (**C**) P2 non-capsid region, and (**D**) P3 non-capsid region and the recombinant lineages are differentiated by distinct colors. The red triangle indicates the CVB5 prototype strain.

**Figure 5 viruses-14-02693-f005:**
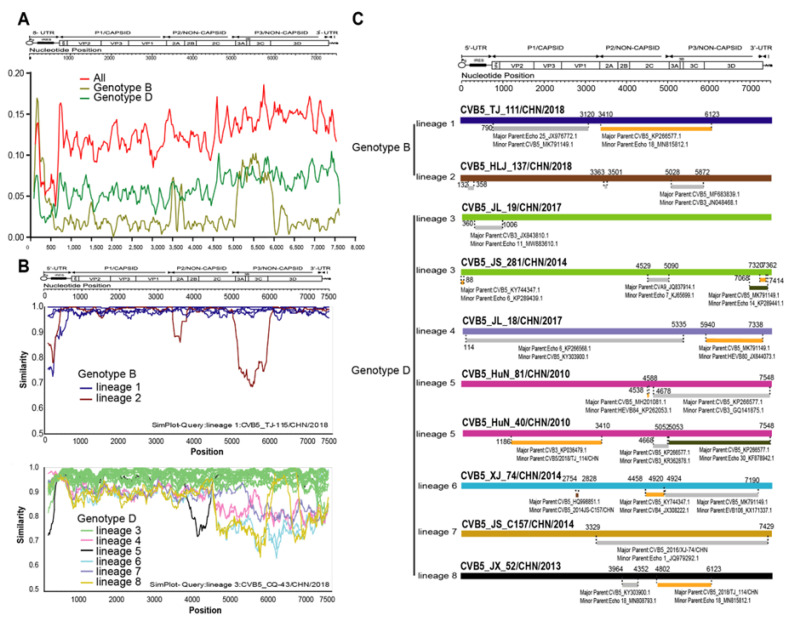
Recombination analysis of CVB5 genotypes. (**A**) Average pairwise diversity based on 53 full-length genome sequences of all genotypes and genotypes B and D. Each lineage is shown using a sliding window of 200nt with a step of 20nt. (**B**) Recombination breakpoints based on the whole genomes of different lineages as detected within genotypes. (**C**) Genomic mapping of representative strain recombination events predicted using RDP4 software for CVB5.

**Figure 6 viruses-14-02693-f006:**
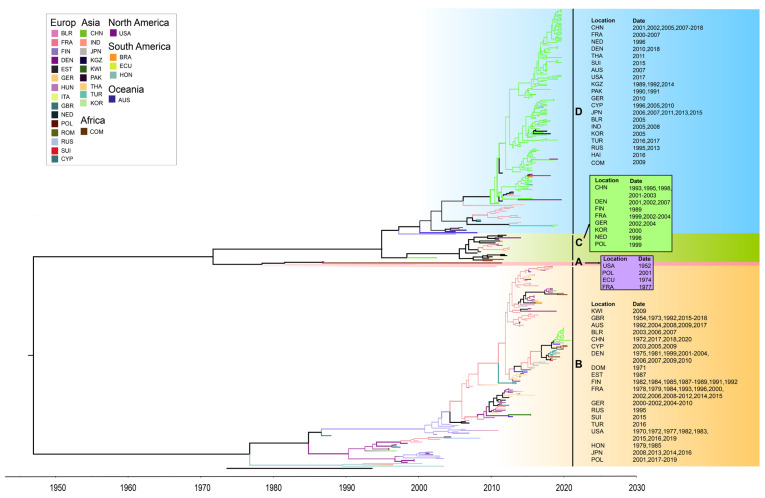
The maximum clade credibility tree of CVB5 worldwide. Geographic locations are abbreviated as follows: BLR: Belarus, FRA: France, FIN: Finland, DEN: Denmark, EST: Estonia, GER: Germany, HUN: Hungary, ITA: Italy, GBR: the United Kingdom, NED: Netherlands, POL: Poland, ROM: Romania, RUS: Russia, SUI: Suisse, CYP: Cyprus, CHN: China, IND: India, JPN: Japan, KGZ: Kyrgyzstan, KWI: Kuwait, PAK: Pakistan, THA: Thailand, TUR: Turkey, KOR: Chesapeake, BRA: Brazil, ECU: Ecuador, HON: Honduras, COM: Comoros, USA: United States, AUS: Australia.

**Figure 7 viruses-14-02693-f007:**
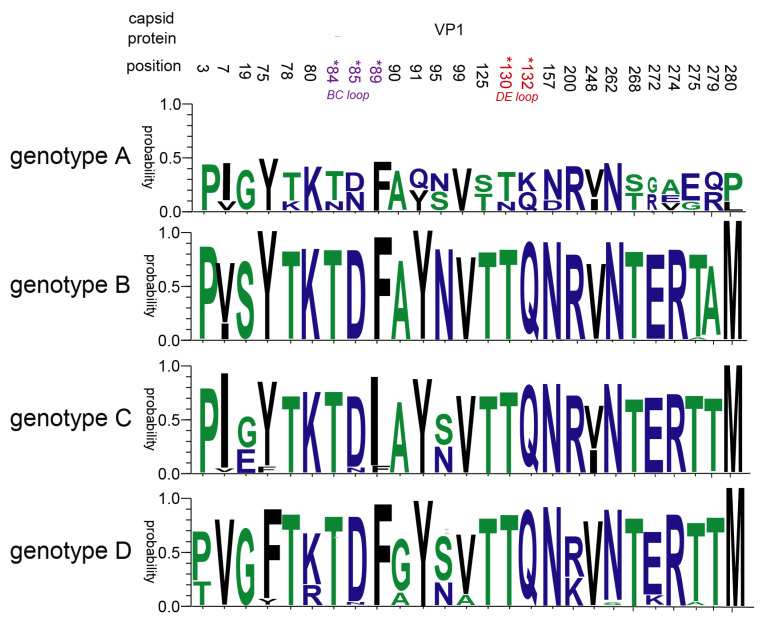
Amino acid mutation in the VP1 region of CVB5.

**Table 1 viruses-14-02693-t001:** Primers used for the full-length genomic sequencing of CVB5.

Primer	Nucleotide Position (nt)	Sequence (5′-3′)	Orientation	Reference
0001S48		GGGGACAAGTTTGTACAAAAAAGCAGGCTTT	Forward	[18]
338F	338–1182	GTTTCGTTCAGCACAACCCC	Forward	This study
1134R	1134–1383	TCAGGCTCTTCTGGTCAGGT	Reverse	This study
1115F	1350–2050	ACCTGACCAGAAGAGCCTGA	Forward	This study
2713R	1949–2968	CCTTCTGTCCAGAACACGCT	Reverse	This study
490	2226–2248	TGIGTIYTITGYRTICCITGGAT	Forward	[17]
491	2883–2902	ATGTAYRTICCICCIGGNGG	Forward	[17]
492	2953–2934	GGRTTIGTIGWYTGCCA	Reverse	[17]
493	3641–3622	TCNACIANICCIGGICCYTC	Reverse	[17]
2183F	2183–3550	CTTCTACCAAGGGCCCACAG	Forward	This study
3942R	3250–3942	GCTTGAGCCTGGTCAGGAAT	Reverse	This study
3350F	3350–4360	AAACTGCGTGTGGGAGGATT	Forward	This study
4450R	3650–4450	GCCCGATTAGATTGGTGGCT	Reverse	This study
3935F	3935–5000	GCTCAAGCAACTCCCACTTC	Forward	This study
4825R	4825–5100	GGGATTTCAGGTGCAACACT	Reverse	This study
4605F	4605–5830	TGTTGCCCAGTCAATTTCAA	Forward	This study
5579R	5579–6730	TACCTTGCCGGTACACATGA	Reverse	This study
5600F	5600–6580	GACACTGCTGAAACTCAACCG	Forward	This study
6500R	6500–6880	GTCTGGATTTCCCTTTTGCCAC	Reverse	This study
5991F	5991–6730	TTGACAACAAGTGCGGGATA	Forward	This study
6900R	6240–6900	CGTGGTCTTGGGTGTTCTTT	Reverse	This study
6758F	6758–6940	CTTTGTGCGTGGTGGTATGC	Forward	This study
7380R	6900–7380	CCCTACCGCACCGTTATCAA	Reverse	This study
7500A		GGGGACCACTTTGTACAAGAAAGCTGGG(T)_24_	Reverse	[19]

## Data Availability

The full-length genomic sequences for the ten strains determined in this study have been deposited in the GenBank nucleotide sequence database under accession number OP672300-OP672309.

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
