# Peer review of "Molecular Epidemiology Reveals the Co-Circulation of Two Genotypes of Coxsackievirus B5 in China"

_viruses, 2022, doi:10.3390/v14122693_

Round 1

Reviewer 1 Report

My comments

Line 2 – Even though the group performed the complete sequencing of 53 viruses and had a high sampling of complete VP1 sequences in my interpretation is better say “Molecular epidemiology” than “Genomic epidemiology”.

Line 36 – Instead of the word “reveled”, I think the expression “showed some important aspects of” is more appropriate.

Lines 69-70 – all positive samples isolated in RD and Hep-2? Only in RD or only in Hep-2? It is not clear for me. There were differences in genotypes isolates?

Line 133 – only one sample from southwest China. Is there any reason for that?  It is important to discuss this point. If no reason its necessary to say that more data from this region is necessary to clarify.

Lines 198-201 – More current studies are needed to support this hypothesis (citations are from 1987 and 2006). Its important to say that in the text.

Lines 310-312 - Is there any reason that can be discussed regarding the result found for the northeast region to have a high rate of migration events? Any important economic, social, cultural characteristics to be highlighted?

Reviewer 2 Report

The manuscript “Genomic Epidemiology Reveals the Co-circulation of Two Genotypes of Coxsackievirus B5 in China describes the spatiotemporal dynamics of an important enterovirus neurotropic (CVB5) associated with HFMD. Additionally, the authors carried out a recombination analysis between two principal circulating CVB5 genotypes in the Chinese mainland. HFMD is a common self-limiting childhood infection affecting particularly those under 5 years old. It is recognized to be caused mainly by enteroviruses. This manuscript can be of broad interest to those that investigate the genetic characteristics and the CVB5 evolutionary process. I have significant concerns with the manuscript and suggest that it is not suitable for publication as it is now. However, I can reconsider my decision if the recommendations below will be addressed.

1.      English needs to be carefully revised.

2.      In the context of global eradication of polioviruses, it is important to highlight the possible role of CVB5 associated with AFP cases.

3.      Please change serotype for type

4.      Please provide the specimen types that CVB5 was isolated.

5.      Please include in the abstract that the analyses were performed with 69 entire VP1 coding regions.

6.      Which parameters were defined to classify HFMD (severe or mild) cases?

7.      In line 184, the authors mentioned that “…the frequency of genetic variation suddenly decreased and gradually reached a plateau after 2018, suggesting that the number of infected hosts increased significantly and showed an exponential trend during 2015-2018.”(lines 184-187) The statement seems contradictory. The genetic variation in the CVB5 VP1 region is proportional to the number of infected hosts. Could you explain that? Describe your findings in the context of the effective population size (EPS).

8.      Supplementary table 3 was not available to the reviewers. Furthermore, is there a supplementary table 2?

9.      The authors conducted the study with HFMD samples (except for one aseptic meningitis sample) and suggested that genotype C circulated from 1970-2011. However, an outbreak of aseptic meningitis caused by CVB5 genotype C that circulated from 2002 to 2012 occurred in China (Shen et al., 2018). Given that the study is based on HFMD samples, it is challenging to make such a claim. Furthermore, I suggest the authors reconsider maintaining the title as written. I propose something along the lines of “Genomic Epidemiology Reveals the Co-circulation of Two 2 Genotypes of Coxsackievirus B5 in China associated with HFMD cases”.

10.  The substitution rate (s/s/y) for full VP1 is different compared to full-length genome sequences. Additionally, it’s widely known that the VP1 region is less variable than the non-capsid region.  Explain why VP1 was used in temporal phylogenetic (MCC) studies instead of complete genome sequences.

11.  The references must be updated. In recent years, numerous reports have discussed the molecular evolution and genetic diversity of CVB5.

12.  I believe the results can also be explored more thoroughly if the authors provided a graphic depiction of sequence variations in the VP1 region using the WebLog program. 

13.  Please, the EV nomenclature must be according to the Recommendations for the nomenclature of enteroviruses and rhinoviruses (Simmonds et al., 2020). Please, check it.

14.  Line 323: Echol 80? Did you mean EV-B80?

15.  Is there a correlation between case severity (mild or severe) and circulating genotypes?

16.   Provide more details about the evolutionary pressure measures (dN/dS ratio).  Please verify your findings to dN/dS value according to dN/dS<1 as negative selection, dN/dS = 1 as neutrality, and dN/dS>1 as positive selection.

         17.  Please provide a paragraph describing the limitations of this                    study.

Round 2

Reviewer 2 Report

 The authors have significantly improved the manuscript. However, I still have some minor issues:

1.      Please, the EV nomenclature must be according to the Recommendations for the nomenclature of enteroviruses and rhinoviruses (Simmonds et al., 2020). Lines 340, 341

2.      Supplementary table 1: Kindly submit your sequences generated to GenBank and include accession numbers in the manuscript.

Author Response

Dear, reviwer
